# The Impact of Sustainability Goals on Productivity Growth: The Moderating Role of Global Warming

**DOI:** 10.3390/ijerph182111034

**Published:** 2021-10-20

**Authors:** Domicián Máté, Adam Novotny, Daniel Francois Meyer

**Affiliations:** 1Department of Engineering Management and Entrepreneurship, Faculty of Engineering, University of Debrecen, H-4028 Debrecen, Hungary; 2College of Business and Economics, University of Johannesburg, Johannesburg 2006, South Africa; novotny.adam@uni-eszterhazy.hu (A.N.); dfmeyer@uj.ac.za (D.F.M.); 3Faculty of Economics and Social Sciences, Eszterházy Károly Catholic University, H-3300 Eger, Hungary; 4Business School, Nord University, 8026 Bodø, Norway

**Keywords:** Sustainable Development Goals (SDGs), productivity growth, climate change, global warming, rising temperatures, CO_2_ emissions, food security, life expectancy

## Abstract

The objective of this paper was to gain novel insights into the complex relationships among Sustainable Development Goals (SDGs) in shaping productivity (GDP/capita) growth. Using dynamic panel regressions on data collected in 138 countries between 2000 and 2017, we found that rising temperatures negatively affect growth and mitigate the impact of other SDGs on growth. We also found that CO_2_ emissions have a U-shaped relationship with growth; life expectancy negatively influences growth (positively moderated by rising temperatures), and food security positively impacts growth (negatively moderated by rising temperatures). This study highlights the difficulty of simultaneously implementing SDGs and elucidates novel research perspectives and policies to decrease the negative impacts of climate change on socio-economic and environmental well-being.

## 1. Introduction

Following the Millennium Development Goals (MDGs), the Sustainable Development Goals (SDGs) were introduced in 2015 to emphasise global efforts toward reconciling economic and social with ecological aspirations [1]. The 17 SDGs comprise 169 targets related to poverty, hunger, health, education, gender equality, water, energy, work and growth, industries, inequality, communities, consumption, climate, oceans, biodiversity, institutions, and international partnerships [2]. Since development goals and targets depend on and influence one another [3], implementing them simultaneously in a coherent manner is a daunting task policymakers have to face. It is unclear how these interlinkages work, or how progress on one goal or target influences other goals and targets through causal relationships and feedback loops [4].

Mitigating the impact of global warming is vital for the future of humanity. In 2019, the earth’s surface temperature was around 0.95 degrees Celsius (°C) warmer than the 20th-century average [5]. Temperatures have consistently been among the hottest for years, increasing sea levels and decreasing Arctic ice [6]. As a result of increasing global surface temperatures, weather-related disasters have become much more frequent, and the number of extreme events is increasing yearly [7]. Droughts, storms, and floods caused catastrophic damages worldwide and resulted in almost $129 billion of economic loss in 2016 [8].

Rising temperature is likely to have a continued negative effect on societies and economies. Shared socio-economic pathways explore possible paths for climate change projections that could affect a wide range of future trends [9]. Xu et al. [10] forecast that areas inhabited by one-third of the human population could become the hottest parts of the world in 50 years unless greenhouse gas emissions (GHGs) are reduced. Climate change and global warming will lead to the permanent loss of critical resources, droughts and floods, imbalances in ecosystems, extinction of species, and threats to human life [11].

Global warming has direct impacts on human productivity, i.e., output per capita (productivity). Roson and Mensbrugghe [12] assessed various climate change effects (e.g., rising sea levels, variations in crop yields, water availability, human health, tourism, and energy demand) and found that the effect of rising temperatures on real GDP is significant and impacts are especially severe for developing countries. Recent studies have focused on the extent to which temperature change and CO_2_ emissions contribute to per capita growth or total factor productivity (TFP) [13].

This paper aims to better understand the impact of rising temperature on productivity growth by examining its direct and moderating role. How rising temperature interacts with other sustainability challenges in shaping growth has received very scant attention. After examining the impact of food security (SDG2), life expectancy (SDG3), and GHG emissions (SDG13) on productivity growth, we tested novel hypotheses vis-à-vis the moderating effect of rising temperature.

We estimated two-step dynamic panel regressions built on a Cobb Douglas production function, using a sample of 138 United Nations (UN) member states. The advantage of the dynamic approach is to eliminate the deeper lags of the dependent variable, which reduces the number of observations available while also taking endogenous economic growth into account [14].

The remainder of the paper proceeds as follows. In Section 2, we elaborate six hypotheses based on the literature. In Section 3, we present the variables and data analysis method. The results of the regression analyses are presented in Section 4. The final section (Section 5) suggests conclusions and implications for future research and policy.

## 2. Literature and Hypotheses

**Hypothesis 1** **(H1).**
*Global warming (rising surface temperature) has a negative impact on productivity growth.*


SDG13 aims to combat climate change and its impacts by regulating emissions and promoting developments in renewable energy [15]. Our preliminary hypothesis assesses the direct effect of climate change (temperature rise) on growth. It is generally accepted that climate change impacts output and growth substantially, especially in emerging countries and in the long run [16]. Rising temperature influences economic growth directly by reducing agricultural output and crop yields [17], industrial output, and labour [18]. The sensitivity of productivity to climate change could be much higher than predicted by direct damage functions, estimating a 23 per cent decline in global GDP by 2100 [19]. The effects would be overwhelmingly adverse at the end of the century and significantly higher in developing countries [20].

**Hypothesis 2** **(H2).**
*Increasing carbon dioxide (CO_2_) emissions have a negative effect on productivity growth.*


While the direct impact of CO_2_ on temperature is well-documented, the overall effect of emissions is not apparent because of feedbacks and complicated interconnections in the ecosystem [21]. The atmospheric CO_2_ concentration has been seen as a significant contributing factor that causes global warming [22]. It is also widely accepted that economic growth is coupled with increased levels of CO_2_ emissions. However, the exact nature of the relationship between growth and environmental degradation is not straightforward. While much of the literature deals with the effect of economic output on CO_2_ emissions, some scholars found reverse causality running from carbon emissions to growth [23]. Developing economies are even more vulnerable, as they use more emission-intensive technologies, ultimately decreasing their economic growth [24].

The environmental Kuznets curve suggests an inverted U-shaped relationship between income per capita and environmental quality [25]. The environment gradually degrades as countries increase production, but after a certain level of growth and standard of living, societies begin to improve their relationship with the environment. Some other studies [26,27,28] also indicated a robust non-linear relationship between CO_2_ and economic growth as captured by TFP growth. However, the relationship between emissions and economic growth is likely to be affected by the myopia of societies, the ability to implement intergenerational transfers, and the externalisation of pollution over borders [29]. Chavaillaz et al. [30] predicted an additional annual loss of labour productivity of about two per cent of total GDP per unit of trillion tons of carbon emitted. However, some individual countries (e.g., China, Japan, and the USA) show a significantly positive relationship between economic growth and carbon emissions [31].

**Hypothesis 3** **(H3).**
*Increasing life expectancy positively impacts productivity growth.*


SDG3 envisions healthy lives and well-being for all people of all ages. It is generally accepted that higher life expectancy is a good proxy of health associated with economic growth. The subject of academic debate is whether an improvement in life expectancy causes an increase in per capita income. On the one hand, improvements in life expectancy generally lead to faster economic growth [32]. On the other hand, Acemoglu and Johnson [33] found that better health conditions trigger faster population growth, which is expected to have a negative impact on productivity. The direction of the effect may depend on the stage of “demographic transition”, after which individuals’ education and fertility decisions start to depend on life expectancy reducing population growth and increasing productivity [34]. The authors argue that most countries today are close to or have passed the demographic transition. Therefore, the effect of increasing life expectancy on per capita income are positive, on average.

**Hypothesis 4** **(H4).**
*Higher food security (measured by MDER) has a positive impact on productivity growth.*


SDG2 aspires to end hunger, achieve food security and improved nutrition, and promote sustainable agriculture. Global hunger estimates are generally based on such indicators as minimum dietary energy requirement (MDER). MDER is the minimum amount of dietary energy (kcal/capita/day) that can be considered adequate to meet the minimum energy needs with low physical activity. The modernisation of industries has improved food supply, and food intake plays a vital role in increasing labour productivity [35]. At the same time, the demand for certain food products sharply increases with economic growth, posing many challenges for food supply chains and food safety [36]. Proper food intake improves health, and better childhood nutrition raises educational attainment, improving productivity through human capital [37].

**Hypothesis 5a** **(H5a).**
*Global warming alters (negatively moderates) the impact of life expectancy on productivity growth.*


The first four hypotheses evaluate the links between key SDGs (climate, health, and food) and economic growth. In the following two hypotheses, we propose that rising temperature alters the effect of other sustainability goals on growth. The interaction of global warming with health and food safety can manifest through various factors, such as mortality [38], extreme climatic events [39], crime and unrest [40], damage to infrastructure [41], as well as adaptation efforts and the production of more expensive carbon-free energy technologies [42]. The causal link between health (life expectancy) and growth may also depend on the ecological consequences of rising temperature [43]. For example, global warming changes the abundance and habitats of organisms that transmit diseases, i.e., vectors, which can shift the seasonal occurrence of several infectious diseases (e.g., malaria, dengue fever, West Nile virus) and cause them to spread. The adverse health effect of rising temperature is exacerbated by crowding, food, and water scarcity [44], and a much depends on the adaptation skills of public health systems, including vaccines and therapies [45].

**Hypothesis 5b** **(H5b).**
*Global warming alters (negatively moderates) the impact of food security on productivity growth.*


Rising temperatures can alleviate the positive effect of food security, and the impacts fall disproportionately on the poor [46]. Warmer temperatures can increase the speed of insect proliferation, increasing the need for food security measures and crop protection [47]. The stability of whole food systems may be at risk under global warming because of short-term variability in supply, which aggravates food insecurity in areas vulnerable to hunger and undernutrition [18]. Access to food and drinking water can indirectly affect household incomes through damage to health [48].

## 3. Research Design and Methodologies

Table 1 presents the descriptions and sources of the variables. Our dependent variable, productivity growth, is calculated by dividing the natural logarithm of real GDP at constant (2011) prices by labour force. Explanatory variables were collected from various sources such as the Penn World Table (9.1) [49], World Bank Databank [50], and Food and Agricultural Organization (FAO) Database [51].

The examined variables’ descriptive statistics (mean, standard deviation, minimum and maximum values, skewness, and kurtosis), and the pairwise correlation matrix of dependent and independent variables can be found in Appendix A (Table A1 and Table A2). The (LLC) test of stationarity statistics was also applied to a subset of the panel data to examine whether the series contained unit root [52].

We employed an unbalanced panel dataset of 138 UN countries (see Figure 1) for the period between 2000–2017. The selected countries cover about 71.5 per cent of the countries in the world, giving the study overall global representativeness. Time-invariant features of static regression methods in the panel data can cause bias across countries [53]. Generalised method of moment (GMM) approaches are better than fixed-effect regression estimates for analysing panel databases, and dynamic methods used to quantify the impact of climate change on economic growth have recently emerged [54]

Similarly to that employed by Mankiw et al. [55], we assumed a Cobb-Douglas production function (Equation (1)). The notations are standard:(1)$$\;Y_{t}=K_{t}^{\propto }{\left(A_{t}L_{t}\right)}^{1-\propto }$$

The sum of total income [*Y_t_*], physical capital [*K_t_*], and labour [*L_t_*], determined by past accumulation in period t. *α* and (1 − *α*) are the elasticities of capital and labour, and constant returns to scale (0 < *α* < 1) are assumed in this model. The technical efficiency of production is denoted by [*A_t_*], as the residual output, which is not explained by expenditures of labour and capital used in production. Both sides of Equation (1) should be divided by [*L_t_*] to determine output per worker (productivity) [*y_t_*] Equation (2):(2)$$y_{t}=k_{t}{A_{t}}^{1-\propto }$$

The term [*k*] represents the capital/labour ratio as the amount of capital available per unit of labour input. The dynamics of capital intensity [*k**] is equal to the amount of [*s_k_*] and the unit of effective labour (*n* + *δ* + *g*) that needs to be invested in preventing [*k*] from falling [56]. Thus, depreciation [*δ*] and technological change [*g*] are assumed to have little effect on the estimates, resulting in a constant (0.05) increase in employment growth [*n*]. Likewise, the steady-state level of productivity [*y_t_**] corresponds with [*k**] Equation (3):(3)$$y_{t}*=A_{t}{\left(\frac{s_{k}}{n+\delta +g}\right)}^{\propto /\left(1-\propto \right)}$$

In our model specification, the economy will tend to return to long-term equilibrium. The steady-state prediction takes logs (ln) from both sides of Equation (3), and the relationship between the explanatory variables is now linear Equation (4):(4)$$lny_{t}*=lnA_{t}+\frac{\propto }{\left(1-\propto \right)}\left[lns_{k}-\left(n+\delta +g\right)\right]$$

Transform and arrange Equation (4) into a linear formula at country i, and time t Equation (5):(5)$$lny_{i,t}*=\beta_{o}+\beta_{1}lns_{k_{i,t}}-\beta_{2}ln{\left(n+\delta +g\right)}_{i,t}+\beta_{3}lnA_{i,t}+\varepsilon_{i,t}$$
where [*lny_i,t_*] is the dependent variable of GDP per capita at constant prices, *lnsk_i,t_* is the ratio of gross capital formation per GDP, and [*ln(n + δ + g)*] is calculated as the sum of the growth rate of employment. [*lnA_i,t_*] denotes the exogenous rate of TFP (total factor productivity), the model remaining after the capital accumulation. TFP can capture the impact of climate change, emissions, life expectancy, and food security on productivity.

The frequent misunderstandings about the neoclassical model is that it fails to explain the catching-up countries. However, the real explanation for the economic growth needs to be derived from the model, which can be understood as the changes that the economy itself (endogenously) forms [57]. Arellano and Bond [58] proposed a generalised method of moments (GMM) model that uses instrumental variables to resolve the endogeneity problem of inconsistencies. Following this dynamic approach, lagged dependent and predetermined variables are used as exceptional instruments. The number of instruments and the maximum lag of the independent variables are limited to avoid overestimating [59]. The two-step (2SGMM) estimators are preferred to the less efficient one-step ones such as least square (LS) and maximum likelihood (ML) [14]. 2SGMM is less likely to be mis specified, and it is more flexible as it does not impose any restrictions on data distribution [60].

After taking the first differences of the dependent variable, the above Equation (5) was transformed as follows Equation (6):(6)$$\begin{array}{c} \Delta lny_{i,t}*=\beta_{o}+\beta_{1}\Delta lny_{i,t-1}+\beta_{2}lns_{k_{i,t}}-\beta_{3}ln{\left(n+\delta +g\right)}_{i,t}+\beta_{4}Temp_{i,t}\\ \;\;\;\;\;\;\;\;\;\;\;\;\;\;\;\;\;\;\;\;\;\;\;\;\;\;\;\;\;\;\;\;\;\;\;\;\;\;\;\;\;\;\;\;\;\;+\beta_{5}{CO_{2}}_{i,t}+\beta_{6}CO_{2}sq_{i,t}+\beta_{7}+lnLife_{i,t}+\beta_{8}lnMDER_{i,t}+\varepsilon_{i,t} \end{array}$$
where the dependent variable [*y_i,t_*] is the growth ratio of real GDP per capita of the country [*i*] in the period [*t*]. The first independent variable refers to the lagged dependent variable. The second concerns the share of investment in output. [*n*] is the average growth rate of employment, and [*δ*] + [*g*] is assumed to be constant (0.05). [*Temp*] denotes the average temperature change. [*CO_2_*] refers to carbon dioxide emissions. [*CO_2_sq*] is included to test the potential quadratic relationship between productivity and emissions. [*Life*] is life expectancy at birth, and [*MDER*] is the minimum dietary energy requirement.

## 4. Results

Table 2 contains the results of the dynamic regression estimations based on Equation (6). The significant Wald-tests validate the dynamic approaches’ exact choice. Wald-tests imply that a GMM estimator is appropriate in all models, and empirical results can be relied upon for statistical inference [61]. Autocorrelation tests are performed by AR(2) for second-order serial correlations. In all models (1–8), the results demonstrate that all estimators are free from serial correlations and are well-specified. The Sargan tests demonstrate the lack of over-identifying restrictions, and instruments are lower than the number of countries. Therefore, such violations from mean stationarity are not detectable [62]. Assuming economic growth theories, an increase in GFCF as the proxy of the investment rate (s_k_) positively impacts productivity growth, and employment growth (n + δ + g) is negatively related to the dependent variable in both models.

The coefficients of temperature change (Temp) are relatively small and range from −0.005 to −0.072, negatively affecting productivity growth. If the average surface temperature rises from 0 to 1.5 °C, productivity decreases by 0.008 units (0.022 to 0.014), keeping all other variables constant, which is approximately 64 per cent lower than without a temperature rise.

Results also show a U-shaped relationship between productivity growth and CO_2_ emission: a negative relationship between CO_2_ and growth, but a positive one between CO_2_ square and growth (Models 3 and 4). The overall *t*-test (value = 2.11**) also supports the presence of a curvilinear U-shaped relationship. First, as pollution increases, productivity growth decreases (negative relationship) until a local minimum, and afterwards, growth starts to increase again (positive relationship). Models (5–8) indicate that the life expectancy coefficient is significant; however, its sign is negative in all regression models. If life expectancy increases by one unit, GDP per capita will decrease by 0.119–1.197.

Models 7 and 8 show that food security (MDER) positively contributes to GDP per capita growth, while rising temperature negatively affects it. We also found significant two-way interaction effects between temperature change and MDER. Figure 2 and Figure 3 plot these interaction effects; solid and dashed lines indicate significant differences between slopes, based on Dawson [63]. The influence of MDER is more substantial (steeper) at low-temperature change than high-temperature change; hence rising temperature negatively moderates (decreases) the impact of MDER on productivity. Food security has a weaker (positive) impact on growth if global warming increases.

Similarly, we found significant interaction effects between temperature change and life expectancy. Figure 3 shows that both increasing life expectancy and higher temperature change negatively affect productivity growth, i.e., growth is the smallest in countries where both temperature increases and life expectancy are relatively high. More interestingly, global warming seems to mitigate the negative effect of life expectancy on growth, indicated by the difference in slopes. Life expectancy has a weaker (negative) effect on growth if global warming increases.

## 5. Discussion and Conclusions

This study examined how critical sustainable development goals (SDGs) interact in shaping economic growth. We tested the effects of global warming, CO_2_ emissions, life expectancy, and food security on productivity growth and the interaction of increasing temperature with life expectancy and food security. A dynamic panel regression (Arellano and Bond) model estimates multidimensional data with longitudinal properties. This method eliminates the problem of adding deeper lags of the dependent variable, reducing the number of observations available. Contrary to previous approaches, we also considered the moderating effects of global warming, which is necessary for exploring the underestimated relations between socio-economic and environmental challenges.

Our results indicate that global warming negatively affects GDP per capita growth (H1). Carbon dioxide emissions have a U-shaped relationship with productivity growth (H2). In addition to its direct negative impact, global warming also mitigates the effects of other SDGs on growth. While life expectancy negatively influences growth (H3), it is positively moderated by global warming (H5a). Food security positively impacts growth (H4), which is negatively moderated by global warming (H5b). Hence, our data and analysis support H1, H4, and H5b, partly support H2, and reject H3 and H5a.

The adverse effect of increasing temperature on living standards urges policymakers to combat climate change and its devastating impacts worldwide. However, Hasegawa et al. [64] claim that in vulnerable regions such as sub-Saharan Africa and South Asia, implementing stringent climate mitigation policies impacts global hunger and food consumption more adversely than the direct adverse effects of climate change.

The Paris Agreement of 2016 aims to strengthen the global response to the threat of increasing temperature by keeping its increase below 1.5 °C compared to pre-industrial levels [65]. According to our results, if the temperature rises from zero to 1.5 °C, productivity will decrease by 64 per cent. Moreover, the thresholds of heat exposure that will lead to declined labour productivity are likely to be exceeded in warmer parts of the world, which are often developing countries [30]. The most severely affected regions are tropical areas, such as Southeast Asia, North Central Africa, and northern South America.

Contrary to Rigas and Kounetas [24], we found that the relationship between CO_2_ emissions and GDP per capita growth is robustly curvilinear at a global scale. Lower CO_2_ emissions and higher productivity growth are typical of developing (African) countries, while higher pollution couples with higher growth in China, India, and the United States. Several developed countries (e.g., Western European and Scandinavian countries) tend to implement innovative green technologies to accelerate sustainable growth with decreasing levels of GHGs. Meanwhile, countries rich in fossil fuel (e.g., in the Middle East) will also show faster per capita growth with higher environmental degradation [66]. Lower-income countries are more vulnerable to climate change and endure more substantial economic losses than higher-income ones [67].

We found that food security positively influences growth, but life expectancy has a negative impact on it. It appears that most countries have not achieved the demographic transition yet when population growth decreases because of increasing life expectancy. In the long run, increasing life expectancy will contribute to productivity growth due to accelerated human capital accumulation [68].

Results also suggest that besides its direct (negative) effect, rising temperature moderates the economic impact of other SDGs. Global warming can exert influence on other SDGs in many ways as it impacts health, food and water scarcity, weather conditions, heat exposure, and diseases [42]. SDGs should not be studied in isolation as there are complex interdependencies among them. For example, global warming exacerbates poverty, especially in developing countries, where the incidence of agriculture and other outdoor activities is relatively higher [12]. Climate change negatively affects food quality due to rising temperatures and declining plant growth periods [69]. Global warming also impacts precipitation, which influences soil moisture content and groundwater balance [70]. The effects of long-term climate change, including extreme frosts and sub-optimal temperatures, on the earlier occurrence of flowering and the phenology of (potato) vegetables, transform food distribution and waste undesirably [71]. Food security problems mainly occur in land-based developing countries due to unsustainable arable land usage and irrigation systems. In contrast, land degradation is enhanced by extreme weather conditions such as drought, environmental pollution caused by human activities, and deteriorating soil quality [72].

There are several potential policy implications of the findings. Countries have decided to rebuild their economies; they can only become cleaner, greener, healthier, safer, more resilient, and sustainable by adhering to recovery plans [73]. The post-pandemic recovery requires nations to discover innovative solutions and complex scientific approaches for a more profound, systematic shift towards a more sustainable economy [74]. Climate-positive actions need to trigger the trajectory of atmospheric CO_2_ levels; for instance, green investments accelerate the decarburisation of all aspects of the economy [75]. The availability and accessibility of food, clean water, and better sanitation and hygiene services (WASH) are keys to preserving health and well-being [76]. Rapid progress in reducing hunger and malnutrition over the next decade could pave the way for eradicating extreme poverty through other SDGs [77]. Responses should include making economic recovery packages more resilient to future crises and updating global environmental governance to reverse the degradation of ecosystems worldwide [2].

This study is mainly limited by omitted variable bias, as the variables in the models reflect only a few SDGs (climate, health, food) that we consider vital for the future of humanity. We urge researchers to test further interactions between SDGs (e.g., education, water, energy, innovation, consumption, production, institutions) to help policymakers implement them simultaneously by minimising trade-offs. The Cobb–Douglas production function can also be replaced by a constant elasticity of substitution function or an alternative green growth approach.

Future research also needs to consider theories from various disciplines (e.g., green growth, climate theories) to obtain results and develop global indicators that reflect the complexity of SDGs and their potential future trajectories. Finally, growth should only be supported if it ensures that the people and our planet will continue to provide resources and environmental services for the well-being of all.

## Figures and Tables

**Figure 1 ijerph-18-11034-f001:**
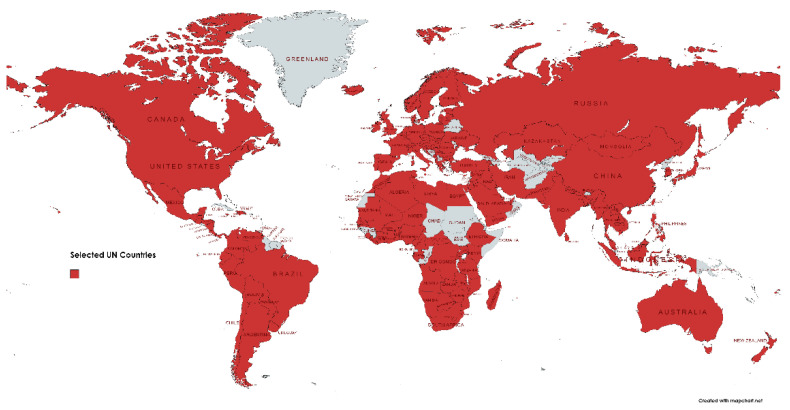
Countries in the sample (selected UN member states are marked with red).

**Figure 2 ijerph-18-11034-f002:**
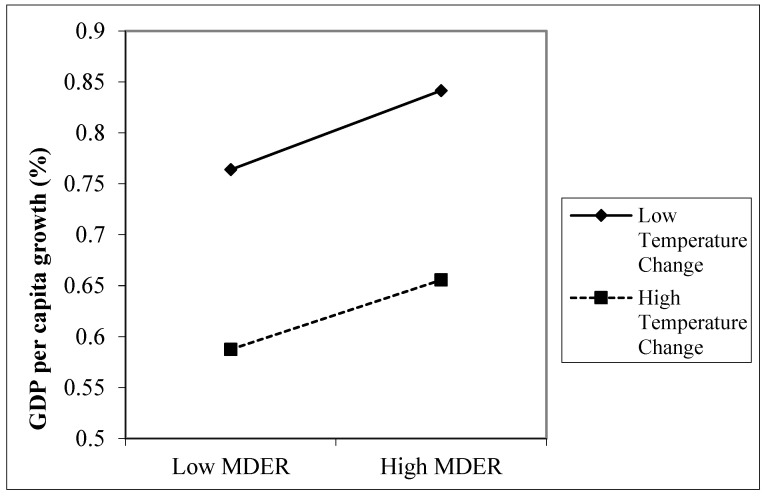
The two-way interaction effects for MDER and moderator (temperature change).

**Figure 3 ijerph-18-11034-f003:**
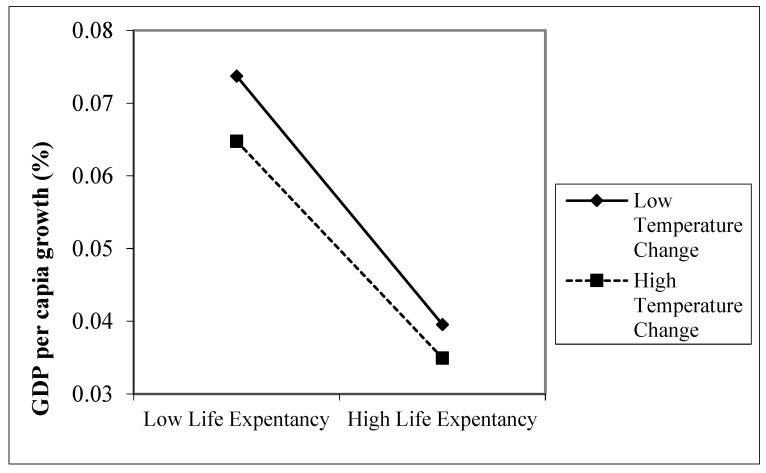
The two-way interaction effects for life expectancy and moderator (temperature change).

**Table 1 ijerph-18-11034-t001:** Description, Abbreviations, and Sources of Examined Variables.

Variable	Abbr.	Description	Source
GDP	Y	Real GDP at constant 2011 national prices (in million 2011 US$)	Penn World Table (9.1)
Gross capital formation (% of GDP)	s_k_	Gross Fixed Capital Formation (GFCF) includes land improvements; purchase of plants, machinery, and equipment; and roads and railways, including schools, offices, hospitals, private residential, commercial, and industrial buildings.	World Development Indicators (WDI), World Bank Databank
Total employment	N	Number of persons engaged (in millions)	Penn World Table (9.1)
Temperature change	Temp	The mean temperature change (°C) range disseminates statistics on the average surface by country, with annual updates.	Food and Agriculture Organisation (FAO)
CO_2_ emissions (kg per 2011 US$ of GDP)	CO_2_	Carbon dioxide emissions stem from burning fossil fuels and cement manufacture during consumption of solid, liquid, and gas fuels and gas flaring.	WDI
Life Expectancy at birth, Total (Years)	Life	Life expectancy at birth indicates how many years a new-born infant would live if the prevailing mortality patterns remained unchanged throughout life at birth.	WDI
Minimum Dietary Energy Requirement (kcal/kg/day)	MDER	The MDER is a crucial factor in malnutrition methodology, as it sets a threshold for estimating the prevalence of an undernourished population in a country.	FAO

Sources: based on [49,50,51].

**Table 2 ijerph-18-11034-t002:** Dynamic panel regression results of Equation (6).

Dependent Variable: Productivity Growth Δln(y)_i,t_								
Independent	Model 1	Model 2	Model 3	Model 4	Model 5	Model 6	Model 7	Model 8
constant	−0.809	−0.072	−0.065	−0.051	0.459	0.582	−4.134	−4.447
	(−12.53) ***	(−1.34)	(−0.96)	(−0.85)	(1.91) *	(3.61) ***	(−4.62) ***	(−5.00) ***
Δln(y)_i,t−1_	0.121	0.112	0.122	0.121	0.109	0.101	0.104	0.106
	(5.66) ***	(5.07) ***	(4.88) ***	(4.56) ***	(4.37) ***	(13.85) ***	(14.43) ***	(14.65) ***
ln(s_k_)_i,t_	0.048	0.048	0.053	0.053	0.053	0.021	0.019	0.022
	(2.69) ***	(2.72) ***	(2.59) ***	(2.84) ***	(2.95) ***	(3.86) ***	(3.71) ***	(3.97) ***
ln(n + g + δ)_i,t_	−0.809	−0.796	−0.807	−0.814	−0.804	−0.781	−0.778	−0.778
	(−12.53) ***	(−11.99) ***	(−12.05) ***	(−12.33) ***	(−12.24) ***	(−40.42) ***	(−39.52) ***	(−39.66) ***
Temp_i,t_		−0.005	−0.005	−0.005	−0.005	−0.068	−0.072	−0.031
		(−2.53) **	(−2.60) ***	(−2.56) **	(−2.55) **	(−1.93) *	(−2.03) **	(−2.38) **
CO_2i,t_			−0.051	−0.103	−0.106	−0.114	−0.111	−0.116
			(−1.63) *	(−1.63) *	(−2.61) ***	(−9.38) ***	(−9.01) ***	(−9.36) ***
CO_2_sq_i,t_				0.016	0.017	0.021	0.019	0.021
				(2.56) ***	(2.67) ***	(10.76) ***	(10.29) ***	(10.90) ***
ln(Life)_i,t_					−0.119	−0.123	−0.175	−0.197
					(−1.95) *	(−3.25) ***	(−4.49) ***	(−4.92) ***
Temp*ln(Life)_i,t_						0.014	0.015	0.041
						(1.76) *	(1.87) *	(3.17) ***
ln(MDER)_i,t_							0.657	0.711
							(5.27) ***	(5.68) ***
Temp*ln(MDER)_i,t_								−0.081
								(−2.81) ***
Observations	1787	1787	1787	1787	1787	1787	1787	1787
Instruments	18	19	19	20	21	68	69	70
Wald test	190.02 ***	207.58 ***	216.64 ***	240.75 ***	240.75 ***	2027.3 ***	1954.4 ***	2104.1 ***
AR(2) test	−0.082	−0.137	−0.147	−0.171	−0.244	−0.211	−0.166	−0.205
Sargan test	31.29 ***	31.28 ***	31.06 ***	31.23 ***	31.24 ***	86.77 ***	87.26 ***	87.41 ***

Note: z statistics are in parenthesis, *** *p* < 0.01, ** *p* < 0.05, * *p* < 0.1.

## Data Availability

Publicly available datasets were analysed in this study. Data can be downloaded from: https://www.rug.nl/ggdc/productivity/pwt/, accessed on 27 September 2021; http://databank.worldbank.org/data/, accessed on 27 September 2021; http://www.fao.org/faostat/en/#data, accessed on 27 September 2021.

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
