# Peer review of "The Impact of Sustainability Goals on Productivity Growth: The Moderating Role of Global Warming"

_ijerph, 2021, doi:10.3390/ijerph182111034_

Round 1

Reviewer 1 Report

General

I believe that this is an interest manuscript on some of the effects of climate change over GDP. My main problem is the hypotheses planted by the authors because they have been dept discussed in many papers, so they are obvious… Even, the authors employed bibliographies that confirmed this. Therefore, their results are not novels according to the current manuscript focus (the hypotheses).

On the other hand, the method or proceeding employed for them could be more interesting to discuss the results from this manuscript, but with focus in comparison with other estimation or evaluation methods or about the integration of SDGs, the impact of this in the world projections under climate change scenarios or general results about MDER, life expectancy, etc.

In this way, the materials and methods section must be improved. To add information about the method selection and the statistical proceeding. The reader doesn’t necessarily know about the Wald, Sargan or Ar2 test and why these methods were selected in this paper. In my opinion, this must be the focus in this paper.

Finally, I recommend to the authors to improve the manuscript and to submit again with this different focus.

Particular

  1. Lines 71-75 are not introduction section. These lines correspond to conclusion section.
  2. Lines 105-107. To clarify that idea. At the beginning of these sentences the authors said: “CO2 contributes to economic growth”, but in the following sentence said: “The impact of CO2 emission on growth is generally negative”. So, to indicate if this impact is negative or positive, because I felt that these sentences are opposed. Moreover, I would like to see some discussion of these sentences with Kuznets curve, especially because this curve showed an increase in GDP in function to increase of CO2 emission until certain level. This must be clarified.
  3. Reorganize the hypothesis: Hypothesis 2, 3, 4, 5a and 5b are clearly described before to give us, but the context for the Hypothesis 1 it is not clear. On the other hand, I believe that is better to put the hypothesis title before to description… This can improve readability.
  4. I don’t understand where CO2sq came from, and why it was used in the models.
  5. The utilization of Chavaillaz et al. (2019) as contrary to Kuznets curve is wrong because that paper is based in projection sceneries, hence their comparison is made in the worst sceneries (RCP4.5 and 8.5), and normally this kind of studies working linearly. In this manuscript, the authors employed historical data and, in this way, it is possible to evaluate the evolution of the emissions and its behavior.

Author Response

Thanks for the opportunity to resubmit a revised draft of the manuscript titled "The Impact of Sustainability Goals on Productivity Growth: The Moderating Role of Global Warming" to the MDPI journal of International Journal of Environmental Research. We appreciate the time and effort that the anonymous reviewers have dedicated to providing valuable feedback on this manuscript. The authors are grateful to the reviewers for their insightful comments on the paper. We have been able to incorporate changes to reflect most of the suggestions provided by the reviewers. We have also highlighted the track changes within the manuscript. We have revised and proofread the Introduction, Literature and hypotheses, Research design and methodologies, and Discussion and conclusion sections. The new version of the manuscript also has updated references, additional 10 high-ranked items are added, and shortcomings are solved. Here is a point-by-point response to the reviewer comments and concerns.

Reviewer 2 Report

The paper explores the direct effects of various SDGs (zero hunger, 61 good health and well-being, climate action) and productivity growth.  The paper uses data from 138 UN member state from the period 2000-2017. The paper is written in a good level of English, there are only a few misspellings.

Major comments:

I do not recommend putting results (p.2. l. 71-74,) in the introduction section. The introduction should describe the relevance of the paper, it should not include the findings.

I wouldn't recommend to list all 138 UN countries, from which the data was collected at the bottom of page 2 since it is hard to comprehend. Instead of a list I'd recommend to put a World Map as a figure, where the involved countries are marked (e.g. have a darker colour). This way the territorial distribution can be better shown.

Figures 2 and 3 should be reformatted. In this presentation there is no indication of the "scale" of the difference betweem the presented values e.g. the difference of 0.55 and 0.75 seems significant, however if it is plotted when the y axis is scaled from 0 to 1 the difference is seemingly much lower. Also the low and high MDER values should not be connected by lines, since the values are discreet. The same can be told for the Life expentancies. The GDP growth doesn't have a unit, which should be added. 

The appendix should be placed after the list of references.

Minor comments:

On page 2 line 53 the ° sign from the °C is not finalized, also a space is missing before it.

I'm not sure why all equal signs (=) are marked with yellow for the equations, but they should be not highlighted. 

Figure 1. is poor quality, it should be improved. The axes should clearly indicate the values presented (name) and the unit of the values (e.g. [°C])

Author Response

(The authors gave the same response as above.)

Reviewer 3 Report

Dear Authors,

Thank you for submitting the article for review in journal “International Journal of Environmental Research and Public Health” and inviting me to a review position. The article is interesting but requires modifications, as it has not avoided minor drawbacks. The authors made a lot of effort in their research, which in some way eliminates the shortcomings. However, the condition for my assessment is the obligatory improvement of the literature review in the indicated direction.

I recommend a weak revision of the article, but not only to reformulate its structure, but to strengthen certain areas.

  1. The introduction does not describe the structure of the article, purpose and hypotheses. In one sentence, the authors basically briefly described the order of writing the article, but these threads should be expanded. The content of the "Literature and hypotheses" section should be included in the introduction, and the review should be devoted to a literature review: systematic or even narrative. The introduction can be divided into subheadings: description of the research problem and article organization.
  2. The literature review did not refer to the results of research by other authors, which have already been widely described and well-established in the literature, as well as indications of the research gap and new developments brought by the research in the article sent to IJERPH. I propose to present the results of other studies in a tabular form, indicate the methods that other authors used, and what their purpose and results were. Discuss what is the gap in this research and how this gap is filled by your research.
  3. Hypothesis 5a should be clarified. Temperature of what? There can be no guesswork in the hypothesis. Hypotheses 1-5b are formulated as conclusions, not as hypotheses, they should be reformulated. As such, they are not falsifiable. You can intuitively guess what the authors would like to emphasize.
  4. The authors refer to the Cobb-Douglas production function, but the model they present is not this function. This is a completely different feature that adapts the Cobb-Douglas approach. Generally, it is before using GDP for this function, it should be decomposed and cleaned, because the interpretation of this function is about potential GDP (GDP*), not real. Further, the calculations do not take into account the correction of the potential work (L*) by NAWRU. Using your abbreviation L*=Nt (1-NAWRU). I propose, in a footnote, to mention that the authors are aware of the necessity to take into account the corrections, but due to some premises (enter what), it was not carried out (these are the so-called limitations of the analysis).
  5. The authors omitted the presentation of information statistics for the comparison of models 1-8. It is recommended to add them and to interpret the best (for information value) model.

I hope that the authors will deal with the introduction of corrections immediately. I keep my fingers crossed, hoping for a positive effect of the work.

Best regards,

Reviewer

Author Response

(The authors gave the same response as above.)

Reviewer 4 Report

The paper is interesting and appealing both considering the theme addressed and the approach chosen to obtain useful results. It is well written and structured, and offers an easy reading throughout the almost whole document.

The references used are up-to- date and consistent with the contents of the paper, and conclusions could be very interesting for the policy implications. However, some parts of the conceptual development are not very clear at a first reading (H2 and H3) and the construction of some hypotheses seems to be based on contradictory reasoning. For example, the formulation of hypothesis 2 is not clear in consideration of the conceptual assumptions presented that say a thing and its exact opposite. In the case of finding discordant positions in the literature, one should declare one's position, justifying it.

The study aims to investigate the relationship between sustainable development goals and productivity growth, considering the moderating role of global warming. There are currently 17 SDGs. This study takes into consideration 3 of them (specifically n. 2, 3 and 13). It is advisable to present the selected goals and argue because the focus is on these three.

With regard to the approach and the methodological rigor, the study has an excellent setting. The sample, the measurement variables and sources of data are clearly presented, as well as the analysis procedure and the results obtained. The authors are asked to explain the choice of placing the study in the perspective of neoclassical theories of economic growth, rather than in contemporary ones (endogenous and institutional evolutionary or green). This choice must also be discussed in terms of the validity (or limits) of the results obtained and the impact on the implications.

Overall, the study is part of a very 'hot' line of studies and offers very interesting, important and urgent considerations for the global socio-economic context.

Author Response

(The authors gave the same response as above.)

Round 2

Reviewer 1 Report

General

Although I still have some doubts about the novelty of the manuscript, especially related with hypotheses, the authors made an effort to improve it.

On the other hand, the quantity of references is excessive for this kind of manuscript (94), normally articles in IJERPH have 50 references, while review have 84, hence I believe that it is necessary to evaluate the real relevance of some of them. Anyway, I believe that Hauer and Santos-Lozada (2021) would be relevant for H5a (Hauer and Santos-Lozada. 2021. Population Research and Policy Review 40: 629-638).

The introduction section described the manuscript idea more than the research context. Although this was improved in comparison with the previous version, it still necessary to eliminate or modify some lines, e.g. lines 36-37; lines 60-61; lines 69-74 and lines 77-81. If the authors would like to explain some research assumptions, that information must added to the methodology section.

Particular

Fig. 1. As recommendation, it must avoid to add figures or tables in the introduction section. This figure must go in the methodology section. This figure must be included in line 183.

Maybe, description of SDG2, SDG3 and SDG13 could add only when they were mentioned in the Hypotheses, and not in the introduction.

Line 153 said “Diversion and loss of resources” What did the authors refer with diversion in this sentence?

Some explanatory sentences in the methodology section are unnecessary. To remember that methodology must be direct and concise. e.g. lines 186-188.

I don’t understand the references as footer in Table 2. Moreover, the results section should not include references.

Sentences about COVID-19 are unnecessary, at least that the authors discussed if climate change had some impact over its propagation.

Author Response

Response to Reviewer 1 Comments (Round 2)

Thanks for the opportunity to resubmit a revised draft of the manuscript titled "The Impact of Sustainability Goals on Productivity Growth: The Moderating Role of Global Warming" to the MDPI journal of International Journal of Environmental Research. We appreciate the time and effort that the anonymous reviewers have dedicated to providing valuable feedback on this manuscript. The authors are grateful to the reviewers for their insightful comments on the paper. We have been able to incorporate changes to reflect most of the suggestions provided by the reviewers. We have also highlighted the track changes within the manuscript. We have revised the Introduction, Discussion and conclusion sections. The new version of the manuscript also has been updated with the references. The suggested one is added, others are (18) evaluated and omitted. Here is a point-by-point response to the reviewer comments and concerns.

General

Point 1: Although I still have some doubts about the novelty of the manuscript, especially related with hypotheses, the authors made an effort to improve it.

Response 1: The authors are grateful to the Reviewer for the favorable comments.

Point 2: On the other hand, the quantity of references is excessive for this kind of manuscript (94), normally articles in IJERPH have 50 references, while review have 84, hence I believe that it is necessary to evaluate the real relevance of some of them. Anyway, I believe that Hauer and Santos-Lozada (2021) would be relevant for H5a (Hauer and Santos-Lozada. 2021. Population Research and Policy Review 40: 629-638).

Response 2: The authors are grateful to support the Reviewer’s comments. The suggested reference is added and others (18) are evaluated based on their relevance.

Point 3: The introduction section described the manuscript idea more than the research context. Although this was improved in comparison with the previous version, it still necessary to eliminate or modify some lines, e.g. lines 36-37; lines 60-61; lines 69-74 and lines 77-81. If the authors would like to explain some research assumptions, that information must added to the methodology section.

Response 3: The authors are thankful to the Reviewer for the comments and suggestions. The suggested lines are carefully corrected, and unnecessary sentences are eliminated.

Particular

Point 4: Fig. 1. As recommendation, it must avoid to add figures or tables in the introduction section. This figure must go in the methodology section. This figure must be included in line 183.

Response 4: Agreed. The Authors appreciated the suggestion of the Reviewer. We included the Figure to the suggested paragraph.

Point 5: Maybe, description of SDG2, SDG3 and SDG13 could add only when they were mentioned in the Hypotheses, and not in the introduction.

Response 5: Agreed. The Authors appreciated the comments of the Reviewer. We reformulated and corrected the introduction and hypotheses sections.

Point 6: Line 153 said “Diversion and loss of resources” What did the authors refer with diversion in this sentence?

Response 6: Agreed. The Authors appreciated the comments of the Reviewer. We windrowed the sentence.

Point 7: Some explanatory sentences in the methodology section are unnecessary. To remember that methodology must be direct and concise. e.g. lines 186-188.

Response 7: Agreed. The Authors appreciated the suggestion of the Reviewer. We deleted the sentence.

Point 8: I don’t understand the references as footer in Table 2. Moreover, the results section should not include references.

Response 8: The Authors appreciated the suggestion of the Reviewer. We deleted the references as footer in Table 2. However, some references were placed in the results section as requested by the other reviewers.

Point 9: Sentences about COVID-19 are unnecessary, at least that the authors discussed if climate change had some impact over its propagation.

Response 9: Agreed. The Authors appreciated the suggestion of the Reviewer. The unnecessary sentences are deleted.

The authors sincerely appreciate the comments and suggestions of the Reviewer.

The corresponding Author
